# Agronomic Performance of Heterogeneous Spring Barley Populations Compared with Mixtures of Their Parents and Homogeneous Varieties

**Linda Legzdiņa \*, Māra Bleidere, Dace Piliksere and Indra Ločmele**

Crop Research Department, Institute of Agricultural Resources and Economics, Zinātnes 2,
LV-4126 Priekuļi, Latvia
\* Correspondence: linda.legzdina@arei.lv

**Abstract:** Diversity within a crop of self-pollinating species can provide advantages in sustainable farming, including the ability to adapt to environments. However, few results proving the benefits in various species and climatic conditions are available. Our aim was to find the differences between heterogeneous and homogeneous materials and determine if crossing has advantage over mixing. We compared essential traits of twelve heterogeneous spring barley composite cross populations (CCPs) to those of seven mixtures representing similar genetic backgrounds and five homogeneous varieties in nine organic and three conventional environments over the course of three years. We found significant advantages for heterogeneous materials, particularly CCPs, for yield in organic and stress environments as well as yield stability, N utilisation efficiency, protein content, 1000-grain weight, and net blotch severity and observed positive trends for N uptake efficiency and weed competitiveness. CCPs' advantages over mixtures were for protein content and 1000-grain weight, a nonsignificant yield gain in low-yield and stress environments, higher yield stability, and minor positive trends for net blotch, NUE, and weed competitiveness. We suggest heterogeneous populations as valuable alternatives to uniform varieties for organic and poor cultivation environments. Although multi-component mixtures could provide a performance similar to CCPs, considering the adaptation potential, populations would be more advantageous overall.

**Keywords:** composite cross populations; mixtures; organic farming; yield stability; competitiveness against weeds; nitrogen use efficiency; disease severity; protein; 1000-grain weight

## 1. Introduction

The principle of diversity has been recognised as essential at all levels from soil microorganisms to plant varieties and cropping systems, providing resilience and sustainability [1]. High-input conventional agriculture and its associated breeding goals have facilitated a reduction in crop genetic diversity, which has threatened yields in less favourable and fluctuating environments [2,3]. Geographically broad adaptations of genetically similar crop varieties widely promoted by the "green revolution" have favoured the extensive use of agrochemicals and significantly decreased the diversity of specifically adapted local varieties that have historically been maintained by farmers [4].

The need to increase genetic diversity within the crop field has been recognised since the beginning of this century, with a particular emphasis on organic and low-input farming systems due to of their location-dependent and seasonal environmental instability [5]. Heterogeneous (evolutionary) populations, also called "modern landraces", can offer effective and inexpensive alternatives to pure-line varieties of self-pollinating species, given their potential superiority in terms of resilience and the ability to evolve and continuously adapt to particular environments as a result of natural selection [6–8]. Composite cross populations (CCPs), consisting of bulked progenies derived from diallel crosses between 28 parents of barley, were first described by Harlan and Martini in the early twentieth

century [9]. Additional relevant findings regarding evolutionary material were reported by Suneson during the mid-twentieth century as a new cost-efficient breeding method that could provide maximum adaptability became available [10]. Subsequent investigations into barley CCPs confirmed local adaptation and the positive effects of natural selection [11], huge improvements in disease resistance over multiple generations [12], marked yield and stability increases [13], and advantages over uniform varieties when grown under abiotic stress conditions [14]. More recent studies were initiated at the beginning of this century in several locations in Europe on various aspects of winter wheat CCPs created in the United Kingdom from 9–20 parents for high yield and/or for preparing flour with characteristics suitable for the preparation of high-quality bread [8,15–21].

The superiority of CCPs under stress conditions has been reported [7]. Advantages in yield, yield stability, and yield reliability in wheat CCPs over the mean of parents across variable cropping environments were reported [8], and compared with popular modern varieties, the experimental populations were found to be comparable or superior in terms of yield when grown under organic conditions [16,22,23]. When tested in low-productive organic and low-input environments, diverse evolutionary barley populations were found to have a highly stable yield across different environments in contrast to homogeneous controls [24]. Experiments on a set of wheat evolutionary populations derived from two and three parents in the United States found significantly larger yield stability over diverse precipitation zones than their respective parents, even though the actual yield itself was not necessarily higher; however, this was only in the environments where particular populations were grown for five seasons [25]. Similarly, Bocci et al. did not observe a positive yield effect in Italian bread wheat populations in regions where they did not evolve [22], whereas a study on population varieties developed by participatory breeding in France indicated that several populations were productive while grown under both favourable and unfavourable organic environments and had good stability across seasons, which is essential for farmers [23].

In one study by Brumlop et al. [16], no decrease in the grain quality of wheat CCPs was found over time, though the stabilising effect of diversity on the grain quality parameters of wheat populations was shown in other studies [19,26]. Döring et al. reported on the superiority of CCPs regarding plant traits that ensure competitiveness against weeds compared to monocultures with the same genetic background [8]. Compared with modern varieties, CCPs were found to have root traits that are better adapted for organic environments and, therefore, better able to compete against weeds and use nutrients efficiently [15]. Evolutionary populations have been recommended as an ecological solution for disease control [22]. In environments where diseases are present, plants with disease resistance prevail [27]. Some indications of lower disease severity in comparison with the parental genotypes have been found [25]; however, the diversity level was not observed to impact on foliar disease, as noted by Döring et al. [8].

Evidence of positive evolutionary changes that demonstrate the ability of heterogeneous populations to adapt to particular growing environments was reported in several recent studies [22,25,28]. No specific local adaptation of wheat populations to particular farm environments could be clearly proven in [26], and no allele frequency differences in 6–10 generations due to organic and conventional cultivation using simple-sequence repeat (SSR) markers in wheat CCPs [18] and barley simple cross populations [29,30] were found. However, a divergence in the major genes controlling plant height, phenology, and disease resistance was noted [18,29]. Several early vigour-promoting seedling root and shoot traits were significantly improved while cultivating CCPs over five and nine generations under an organic—but not under the conventional—crop management system, suggesting an improvement in nitrogen uptake from deeper soil layers and competitiveness for light, which are essential, particularly for organic production with a limited nutrient supply and the presence of weeds [20,21]. In addition, improved tillering ability was reported in organically cultivated barley populations in contrast to conventionally cultivated populations of the same origin [29]. A study on the effects of natural selection under low-input

management demonstrated the dominance of plants with traits enhancing their competitive abilities, such as higher plant height at the beginning of stem elongation, early heading, and a larger number of fertile tillers [31], whereas Bocci et al. found adaptations for height (i.e., taller) but not for weed suppression or crop ground cover [22].

Choosing adapted parents to create populations for particular environments to increase the benefits gained from natural selection has been emphasised [25], and it appears to be critical for populations with traits that are not directly affected by natural selection, such as grain quality [16]. Genetic background is shown to affect yield stability: a population of high-yield genotypes is more stable under conventional conditions, whereas broader genetic backgrounds provide better stability under organic conditions [32].

Organic heterogeneous populations for all crop species are becoming more accessible for farmers due to legislation coming into force from 2022 [33], and information on agronomic performance and comparisons to varieties grown on farms are useful for practitioners when it comes to promoting the occurrence of genetic diversity within a crop field.

The objective of this study was to compare the essential agronomic traits in spring barley composite cross populations (CCPs) to those of mixtures of their parental genotypes and currently grown homogeneous varieties to determine their suitability to organic farming practices and abiotic stress conditions.

## 2. Materials and Methods

### 2.1. Barley Material

Seven covered (CB) and five hulless (HB) spring barley (*Hordeum vulgare* L.) populations were compared to four CB check varieties, Rubiola, Rasa, Abava, and Anakin, and one HB check variety, Irbe, as well as seven mixtures of parental genotypes; there were 24 subjects of barley in total (Table 1).

**Table 1.** Summary of studied barley material.

| Grain Type | Type of Material | Country of Origin | Number of Accessions | Number of Parents | Generation (2019) |
|---|---|---|---|---|---|
| Covered (CB) | population | Latvia | 6 | 10–32 | $F_3$–$F_7$ |
| | | Denmark | 1 | | |
| | check variety | Latvia | 3 | | |
| | | Denmark | 1 | | |
| | mixture of parents | | 6 | | |
| Hulless (HB) | population | Latvia | 4 | 10–32 | $F_4$–$F_7$ |
| | | Denmark | 1 | | |
| | check variety | Latvia | 1 | | |
| | mixture of parents | | 1 | | |

Detailed information on the material is provided in Table S1. Two of the populations were multi-line mixtures originating from Denmark (A. Borgen and Agrologica), and the rest were locally bred composite cross populations (CCPs). At the start of the experiment (2019), CCPs were in the $F_3$–$F_7$ generations. The number of parents used varied between 10 and 32, including male-sterile parents for three CCPs, which were applied with the aim of making crossing occur more quickly and to add additional diversity arising from those parents as well as potential natural pollination in $F_2$. The majority of parents had two-row spikes, and only a few parents of CCP-5, CCP-6, and CCP-7 were six-row. The male-sterile accessions were originally provided by D. Falk [34] and repeatedly backcrossed to local breeding material. Five CCPs originated from diallel crosses, whereas for those involving male sterility, only one-way crosses were possible. CCP-7 was combined from

selected crosses of other CCPs and, therefore, involved larger numbers of parents but less cross-combinations. The main target traits essential for organic growing (with an emphasis on yield, stability, and resistance to diseases) were defined for each CCP, and the parents were chosen accordingly. Hybrid seeds obtained from individual cross combinations were bulked in roughly equal quantities for multiplication in the $F_1$–$F_2$ generations, with the exception of CCP-6 and CCP-7, where $F_2$ seeds were bulked. The seed of each population was divided into two parts, and multiplication was performed in organic and conventional crop management systems in parallel for 1–5 seasons, depending on the creation time, to be used in the field trials for the respective systems. Hulless barley (HB) seeds were selected out of the $F_2$ generation harvest for populations involving some HB parents (CCP-3, CCP-5, and CCP-7) and cultivated with the denominations CCP-3 HB, CCP-5 HB, and CCP-7 HB. HB descendants were discarded from the initial CB population seed thereafter. In the HB CCPs, threshability was improved by only sowing seeds with naturally detached hulls. CCPs involving male sterility (CCP-3, CCP-6, and CCP-7) were subjected to additional cross-pollination in $F_2$, and starting from $F_3$, a negative selection of sterile seeds was performed to eliminate the sterile plants and their negative effect on grain yield.

The mixtures of parental genotypes used in seven CCPs were created in spring 2019 by combining germinable seeds of all the parents involved (except the sterile parents and three missing exceptions—see Table S1) in equal numbers. Separate mixtures for organic and conventional trials were developed using seeds multiplied in the respective systems in Priekuļi in the previous season.

In 2019, the seeds used for the trials were harvested in 2018 from Priekuļi from both the organic and conventional management systems. In the following two seasons, seeds harvested from each site were used to prepare the seeds of the corresponding system for the next year. The seed rates were 400 and 450 germinable seeds per m$^2$ for CB and HB, respectively.

### 2.2. Experimental Sites and Crop Management

Field trials were arranged during three growing seasons (2019–2021) in two geo-graphically and climatically distinct locations: Priekuļi and Stende. A randomised block experimental design with four replications was applied. In Priekuļi (57°19′ N, 25°20′ E), trials were performed in organically (PrOrg) and conventionally (PrConv) managed fields of a research institute, and in Stende (57°11′ N, 22°32′ E), trials were performed in two organically managed sites, one of them in a research field of the institute (StOrg) and the other on a nearby organic farm (StOrgF) 1–2 km away. The soil characteristics and crop management specifications are summarised in Table S2. The soils of the trial sites were Albeluvisols (AB), except in StOrgF 2020, where it was Umbrisol (UM), with all having a loamy sand or sandy loam texture. The conventional fields were managed with a moderate input of agrochemicals (mineral fertiliser provided to obtain 5 t ha$^{-1}$ yield, one application of herbicides, and no fungicides). The organic fields had been certified for organic farming for more than three years. No fertilisers were provided at the organic sites, and the nutrient level was maintained by growing green manure crops in rotations. Crop management, with minor differences from the typical farm practice, was applied at the farm site. The trials were conducted between a spring cereal seed production field and intercropped with red clover (*Trifolium pratense* L.) before or after sowing barley. However, the clover plants showed poor development in 2019. Harrowing was performed at the organic institutional sites before the barley tillering (growth stage (GS) 13–21 according to the BBCH-scale) (in PrOrg 2020, also before plant emergence) to restrict weeds. In the conventional sites, herbicides were applied.

### 2.3. Meteorological Conditions

The average air temperature during barley vegetation (April–August) exceeded the long-term norm in both locations during the three seasons by 0–1.9 °C, and the amount of precipitation was 194–360 mm, or 62–115% of the long-term norm. In general, June was

warmer than usual in all years, and July was extremely hot in 2021 (Figure 1), reaching a 23.2 °C average over the first 10-day period in Priekuļi. Extended periods of rainfall were observed in Priekuļi in 2021 at the end of April and the beginning of May (148% above the norm), interrupting sowing and causing a 19-day difference between the sowing dates in the PrOrg and PrConv sites. June was always drier in Stende compared with Priekuļi, with significantly less rainfall in 2021, which enhanced the stress caused by the heat. This was followed by a comparatively wet July in Stende and a dry July in Priekuļi in 2021.

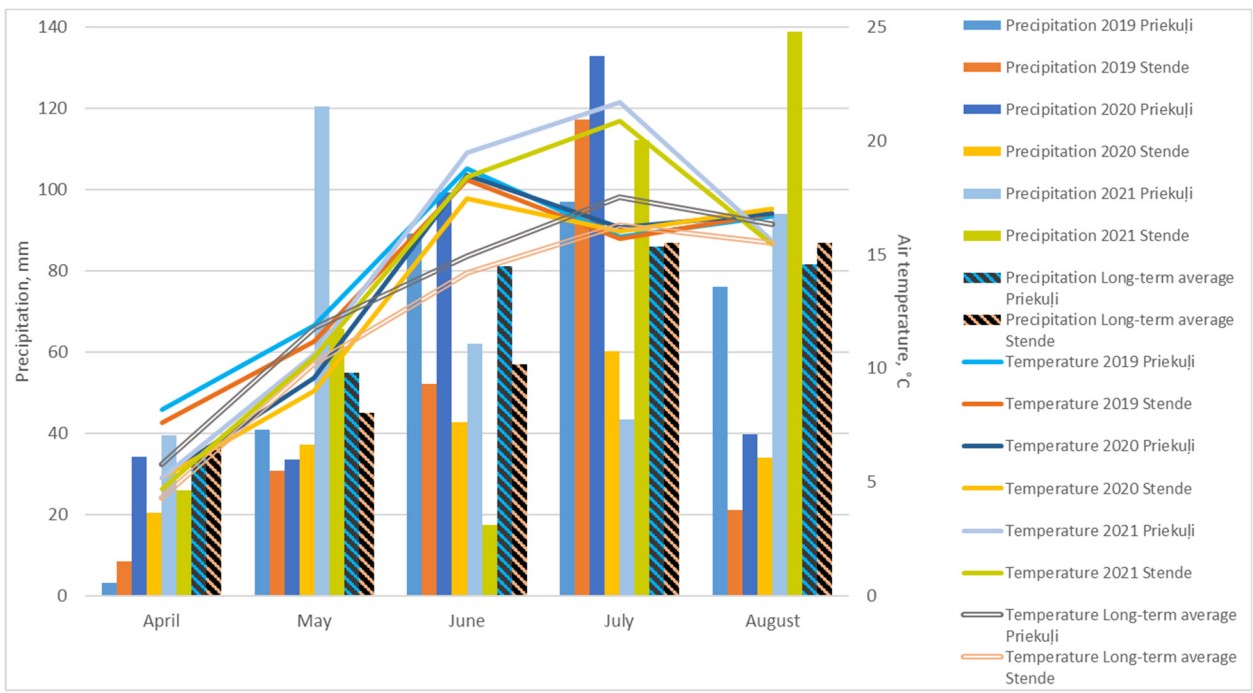

**Figure 1.** Average monthly air temperature and sum of precipitation during barley vegetation in Priekuļi and Stende during 2019–2021.

### 2.4. Assessments and Measurements

The grain was combine-harvested on whole plots, dried, and purified using a 1.7–1.8 mm sieve; the moisture, weight, and yield at 14% moisture in t ha$^{-1}$ were then determined.

To evaluate the weed suppression ability at the organic farming sites during three barley development stages (GS 31–39, GS 59–65, and GS 87–92), a visual assessment of weed ground cover was performed. The weed suppression ability of each genotype was calculated as the difference between the weed ground cover in plots with crop and the maximum weed growth in plots without crop, expressed as a percentage [35]. In addition, crop ground cover was visually assessed at barley GS 25–29 and GS 29–31, as it was found to indicate early competitiveness against weeds [36]. Grain quality characteristics, such as the content of crude protein (Infratec 1241 and Infratec NOVA (Foss, Denmark)) and 1000-grain weight (TGW), were assessed in yield samples from each replicate.

To assess nitrogen use efficiency (NUE, kg kg$^{-1}$, expressed as kg of *grain obtained from* kg N$^{-1}$), aboveground plant biomass samples were collected at two organic sites (PrOrg and StOrg, with the exception of PrOrg in 2020) from a 0.1 m$^2$ area of two rows in the middle of plots at physiological maturity for each replicate. Plant biomass samples were directly weighed, and grains were weighed after being threshed. The number of grains per spike and the spike weight were also calculated. Samples of grain and straw biomass were separately analysed to determine the total N concentration. The dry matter content of grains and plant samples were determined after oven drying at 130 °C for 2 h (ISO 712:2009). NUE was defined according to Moll et al. [37] and computed by multiplying two NUE primary components: N uptake efficiency (NUpE, kg kg$^{-1}$, expressed as kg crop N per kg

soil N) and N utilisation efficiency (NUtE, kg kg$^{-1}$, expressed as kg grain DM per kg crop N). NUpE was calculated as the total amount of N in the aboveground plant at harvest (Nt) divided by the total naturally available N content in the soil at the time of sowing (Ns), and NUtE was calculated as the grain dry weight (G$_0$) divided by Nt. Nt was determined according to Ortiz-Monasterio et al. [38], summing the dry weights of the N concentration in the grains and in the straw biomass. Ns was considered to be the total naturally available N concentration in the upper 0.2 m soil profile, as determined before sowing, and was calculated considering the soil volumetric weight. We did not consider N that could either be available deeper in the soil or that was released by the mineralisation of organic matter during vegetation. The total N concentration in finely milled grain and plant samples as well as in soil samples was determined using the Kjeldahl method (ISO 20483:2013).

Infections with the foliar diseases of powdery mildew caused by *Blumeria graminis* and net blotch caused by *Pyrenophora teres* were scored 0–9 under a natural infection background at Priekuļi (with the exception of PrOrg 2019), where 0 indicates no visible symptoms of disease and 9 indicates that no green tissues of plants are observed. The assessment was conducted repeatedly, starting at the occurrence of the first disease symptoms, with an interval of 6–10 days, depending on the disease progression, until the symptoms were distinguishable. The progress of the disease was described by the size of the area under the disease progress curve (AUDPC) [39]. For seed-borne diseases, the number of plants infected by loose smut (*Ustilago nuda*) and covered smut (*Ustilago hordei*) was recorded in all plots.

### 2.5. Data Analysis

A factorial analysis of variance (ANOVA) and t-tests were performed using a general linear model (GLM) in IBM SPSS Statistics for Windows version 23.0 (IBM Corp., Armonk, NY, USA) to analyse the effects of factors such as year (Y), site (S), genotype (G), environment (E), crop management system (M), grain type (hulless (HB)/covered (CB)), diversity groups, and their interactions (GxS, GxM, GxE, GxY, YxS, and GxYxS) on all the dependent variables (Table S3). For simplicity, the effect of individual subjects (populations, mixtures, and varieties) was designated as an effect of genotype (G) even though heterogeneous materials contain different genotypes. The comparison between diversity groups included: (1) heterogeneous subjects versus homogeneous subjects, (2) populations (CCPs) versus mixtures versus homogeneous checks (separately for CB and HB), and (3) CCPs versus mixtures of the respective parents. In cases where significant differences were observed, the mean values were compared using Tukey's honestly significant difference (HSD) post hoc test at the 0.05 probability level. Pearson's correlation coefficients between all dependent variables were also identified, as were Spearmen's correlation coefficients between subject yields in different sites as well as environments grouped by average yield levels (low: <2 t ha$^{-1}$; medium: 2–4 t ha$^{-1}$; and high: >4 t ha$^{-1}$).

To evaluate yield stability/adaptability, a regression of subject yield to the environmental mean was performed according to Eberhart and Russell [40]. The coefficients of regression (b) and deviation from regression ($s^2_{dj}$) in all 12 environments and in 9 organic environments were used as measures of stability. In addition, calculations were also performed for the three conventional environments; however, these numbers were too small, and b was not significant ($p > 0.05$) for more than half of the subjects, and therefore, they were used in the comparison of trends. Rankings in the top third of each environment for each diversity group were summed up and used as additional information.

## 3. Results

The effect of genotype was significant for all the investigated traits, with the exception of NUpE and the traits characterising competitive ability against weeds, whereas environmental factors (e.g., year, site, individual environments, and crop management system) were significant, with the exception of the effects of site and management on powdery mildew severity (Table S3).

*3.1. Yield*

The average yields differed significantly among all sites, with the highest at conventional farm sites and the lowest at organic farm sites (Table S3). The ranking order differed only in 2021, when heat and drought affected yield formation in Priekuļi, in addition to late sowing at the conventional site, resulting in the highest yield of 3.55 t ha$^{-1}$ at StOrg and the lowest yield of 1.66 t ha$^{-1}$ at PrOrg. The highest average yield was in the most favourable year in terms of yield formation: 2020 in Priekuļi and 2021 in Stende. The amount of rainfall in July showed opposing trends at both locations in those years (Figure 1).

Spearman's rank correlations among the genotype yields at the four sites during the study period were all significant, with the highest coefficients between the organic sites; in addition, correlation coefficients between individual environments grouped according to the average yield level were highest between the low and medium yield levels (Figure S1). This indicates that the ranking order was more similar in the organic and lower-yield environments than in organic vs. conventional and low-yield/high-yield environments. Over the three years, the heterogeneous CB material, mostly CCPs, ranked highest, on average, in organic management (top eight), whereas the two newest and potentially higher yielding check varieties, Anakin and Rubiola, ranked highest under conventional management, followed by the same three top-ranked subjects as in organic farming, CCP-Mirga, MIX Mirga, and CCP-4 (Figure 2). The most extreme changes in ranking order between the crop management systems were for Anakin and Rubiola, which performed better in the conventional system, and CCP-7, MIX-DK, and CCP-5, which performed better in the organic system. After reviewing the rankings in both environments and grouping the results according to yield levels, we found that CCP-7 ranked the highest under low-yield organic conditions (<2 t ha$^{-1}$), medium at yield levels of 2–4 t ha$^{-1}$ (organic and conventional stress conditions), and low at the highest yield level of >4 t ha$^{-1}$ in a conventional management system. When comparing the rankings among the four sites (Figure S2), CCP-7 performed the best at PrOrg but ranked in the middle at the other sites, indicating the possibility of specific adaptability to the site of origin. The Danish population MIX DK always ranked higher in Stende (sites StOrg and StOrgF) than in Priekuļi, indicating that the climatic conditions in Stende may have been similar to those at their site of origin. All CB CCPs ranked higher than the respective mixtures, on average, in low-yield-level environments and in PrOrg environments where they evolved, with the exception of CCP-3, and all CCPs were exclusively found to perform better than their respective mixtures under conventional stress conditions (PrConv 2021, Figure S2).

On average, the yield of heterogeneous CB materials, particularly CCPs, significantly surpassed that of homogeneous checks in the organic crop management system (*p* = 0.039), whereas there were no differences in the conventional system (Table 2). After reviewing the data in individual environments (Figure 3) in organic farming, the average yield of CCPs was highest, followed by mixtures and checks, whereas in conventional farming, the trend was the opposite, with the exception of low-yield stress conditions in 2021. The average yield gain of CB CCPs over checks was 16% in low-yield environments and 10% in medium-yield environments, whereas in high-yield environments, there was a 6% yield loss (for mixtures, 9%, 8%, and 4%, respectively). For HB, the yields of populations and mixtures did not differ from check Irbe in low-yield environments but was inferior under medium- and high-yield conditions (Table S4). When comparing seven pairs of CCPs and mixtures composed of the same (or similar) parents/components, significant differences among yields were not found, and there were no significant differences in individual environments or among pairs (Table S4). However, in 9 out of the 12 environments, there was a yield gain for the CCP group, from 1% to 19%, which was the highest in PrOrg in 2019. This is attributed to PrOrg being the site for which CCPs had been adapted before the experiment as well as the mixtures not undergoing any adaptation prior to the first year of the experiment; however, in the following years, natural selection and adaptation were possible in the mixtures. CCPs typically performed better in terms of yield, as compared to mixtures, in low-yield environments (<2 t ha$^{-1}$) (difference of 8%, on average) as well

as in stressful environments in PrConv in 2021 (by 9%). The difference was minor in environments better suited for higher yield levels. The highest relative yield advantage over the respective mixture was found for CCP-5 in organic farming (16%, on average) and for the most diverse CCP-7 in low-yield environments (26%); no gain was obtained for male-sterile-cross-derived CCP-3 and CCP-6 (−7% and −2%, on average, respectively). This was attributed to the lower yield potential of male-sterile parents.

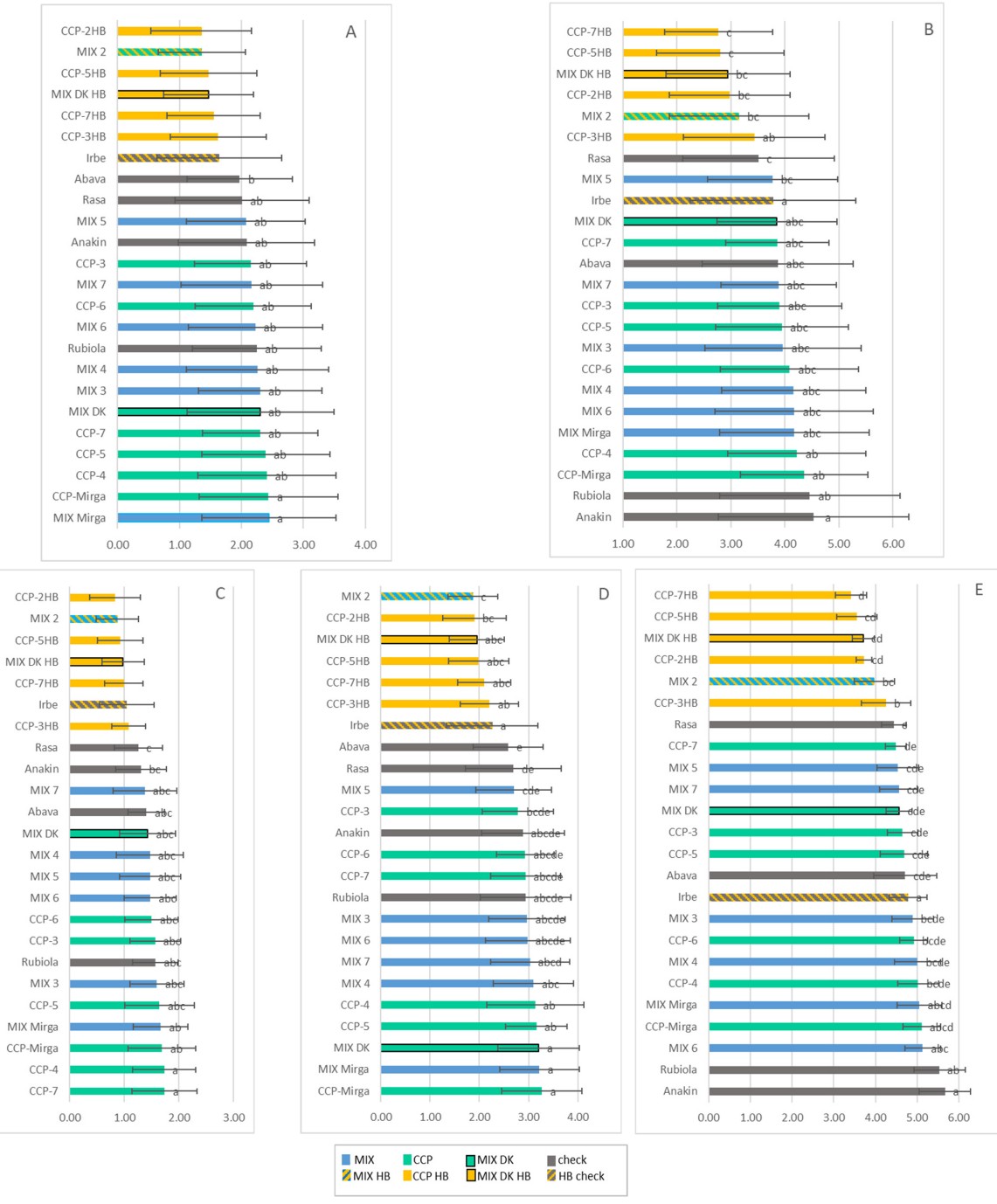

**Figure 2.** Ranking of subjects according to grain yield (t ha$^{-1}$) across all environments: organic, n = 9 (**A**); conventional n = 3 (**B**); low, <2 t ha$^{-1}$, n = 5 (**C**); medium, 2–4 t ha$^{-1}$, n = 5 (**D**); and high, >4 t ha$^{-1}$, n = 2 (**E**). Error bars indicate standard deviations. Values marked with different letters indicate significant differences within CB and HB groups (*p* = 0.05). MIX—mixture, CCP—composite cross population, HB—hulless barley, DK—Denmark (country of origin), check—check variety.

**Table 2.** Significance of differences between subjects grouped by their diversity level (*p*-value) and average values of yield, nitrogen (N) use efficiency (NUE) and its components (N uptake efficiency (NUpE) and N utilisation efficiency (NUtE)), and grain quality traits.

| Diversity Groups | Yield, t ha$^{-1}$ | | | NUE, kg kg$^{-1}$ | NUpE, kg kg$^{-1}$ | NUtE, kg kg$^{-1}$ | Protein, % | | | TGW, g |
|---|---|---|---|---|---|---|---|---|---|---|
| | O+C | O | C | O | O | O | O+C | C | O | O+C |
| **Heterogeneous vs. Homogeneous (CB)** | 0.221 | 0.765 | 0.039 | 0.071 | 0.365 | 0.003 | 0.001 | 0.001 | 0.012 | 0.021 |
| Heterogeneous CB | 2.72 | 4.03 | 2.28 a | 54.77 | 1.40 | 38.79 a | 12.9 a | 14.0 a | 12.5 a | 46.3 a |
| Homogeneous CB | 2.58 | 4.09 | 2.08 b | 50.34 | 1.35 | 36.69 b | 12.5 b | 13.4 b | 12.2 b | 45.6 b |
| **CCP vs. MIX vs. checks (CB)** | 0.419 | 0.955 | 0.093 | 0.128 | 0.444 | 0.010 | 0.004 | 0.002 | 0.021 | 0.022 |
| CCPs CB | 2.74 | 4.03 | 2.31 a | 55.77 | 1.42 | 38.94 a | 12.8 a | 14.1 a | 12.4 a | 46.5 a |
| Mixtures CB | 2.69 | 4.02 | 2.25 ab | 53.59 | 1.37 | 38.62 a | 12.9 a | 13.8 a | 12.6 a | 46.1 ab |
| Checks CB | 2.58 | 4.09 | 2.08 b | 50.34 | 1.35 | 36.69 b | 12.5 b | 13.4 b | 12.2 b | 45.6 b |
| **CCP vs. MIX vs. checks (HB)** | 0.198 | 0.128 | 0.318 | 0.029 | 0.143 | 0.023 | < 0.001 | 0.001 | < 0.001 | < 0.001 |
| CCPs HB | 1.87 | 2.99 | 1.49 | 42.13 ab | 1.31 | 31.97 ab | 14.6 a | 16.3 a | 14.0 a | 41.1 a |
| Mixture HB | 1.81 | 3.15 | 1.36 | 39.23 b | 1.30 | 29.68 b | 15.0 a | 15.9 a | 14.7 a | 39.6 b |
| Check HB | 2.18 | 3.78 | 1.64 | 51.14 a | 1.48 | 34.08 a | 13.4 b | 14.9 b | 12.9 b | 38.4 b |
| **CCP vs. MIX (pairs)** | 0.654 | 0.953 | 0.536 | 0.752 | 0.672 | 0.913 | 0.570 | 0.013 | 0.601 | 0.019 |
| CCPs | 2.61 | 3.91 | 2.18 | 52.26 | 1.38 | 37.26 | 13.2 | 14.6 a | 12.8 | 45.9 a |
| Mixtures | 2.56 | 3.90 | 2.12 | 51.54 | 1.36 | 37.34 | 13.2 | 14.1 b | 12.9 | 45.1 b |

O—organic, C—conventional, O+C—both farming systems; Bold—the row shows *p*-values of comparison between the respective groups; trait values marked with different letters indicate significant differences between the groups (*p* = 0.05); CB—covered barley, HB—hulless barley; NUE expressed as kg grain DM per kg soil N; NUpE expressed as kg crop N per kg soil N; NUtE expressed in kg grain DM per kg crop N.

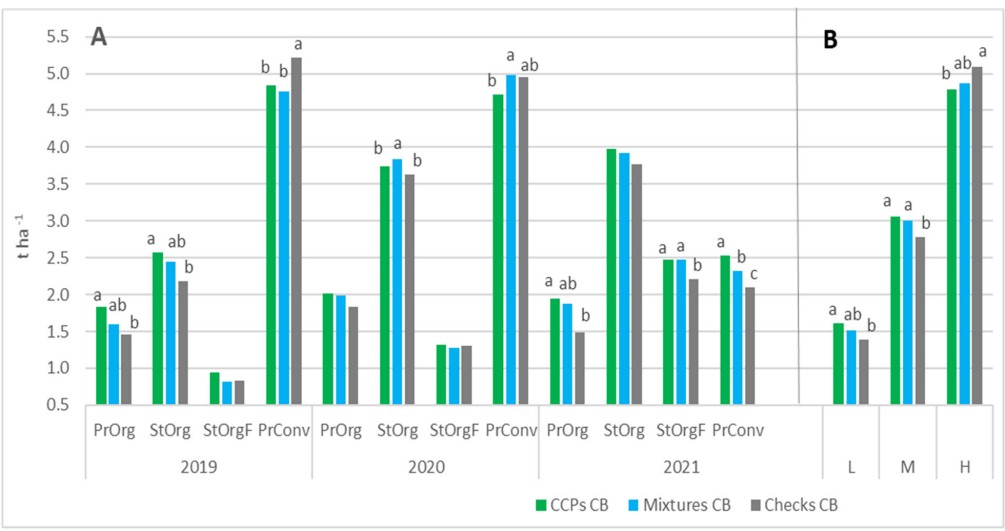

**Figure 3.** Average grain yield of covered barley (CB) composite cross populations (CCPs), mixtures, and check varieties in (**A**) individual environments (organic sites PrOrg, StOrg, and StOrgF and conventional site PrConv during 2019–2021) and (**B**) environments grouped by average yield level (low, <2 t ha$^{-1}$, n = 5 (L); medium, 2–4 t ha$^{-1}$, n = 5 (M); and high, >4 t ha$^{-1}$, n = 2 (H)). Values marked with different lowercase letters indicate significant differences within environment or environment group (*p* = 0.05).

### 3.2. Yield Stability/Adaptability

After reviewing the average values of the regression coefficient (b) and the deviation from regression ($s^2_{dj}$) for the diversity groups (Table S5), we observed a trend whereby the homogeneous checks had the most instability in terms of adaptability to more productive conditions, whereas the CCPs and mixtures were equally stable but with minor polarisation for adapting to less and more productive environments, respectively. All covered barley (CB) CCPs were stable across the 12 environments (b = 1, $p = 0.05$), with the exception of the most diverse population, CCP-7, which showed an adaptability to unfavourable environments ensuring lower yield levels (b < 1, $p = 0.05$) (Table 3). However, CCP-7 was stable over the nine organic trials, indicating its specific adaptability to organic crop management. The CCP Mirga and the CB population from Denmark (MIX DK) were stable across all environments in both crop management systems and provided adaptability to more productive organic environments. Most of the hulless barley (HB) CCPs were unstable and adapted to less productive conditions. The yields of three CB mixtures were stable, and the other three showed adaptability to more productive conditions (b > 1, $p = 0.05$). The HB mixture was stable compared with the respective CCP-2, which showed adaptability to less productive conditions. To compare the b values of the CCPs with their respective mixtures across all environments, five out of seven pairs were lower, whereas the deviation from regression was lower for only two pairs. The most significant differences in b values with the most explicit crossover of regression lines were found in CCP-7 and MIX 7, and to a lesser extent, in CCP-6/MIX 6 and CCP-3/MIX 3 (Figure S3). The two CB check varieties with the highest yield potential (i.e., Rubiola and Anakin) and the HB check Irbe were found to be adapted to favourable environments. We conclude that Rubiola shows higher adaptability to optimum conventional farming conditions since it was stable across the organic environments when analysed separately. When we compared the averaged rankings in the top third across all environments, it was higher for heterogeneous than for homogenous materials, with the differences being smaller in conventional environments; however, for CCPs, it was slightly higher than for the respective mixtures in organic management but similar in conventional management (Table S5). The most stable in terms of yield in the top third was MIX Mirga, closely followed by CCP-Mirga (Table 3).

### 3.3. Protein and 1000-Grain Weight (TGW)

The grain protein content was significantly lower under organic than conventional management (12.9% and 14.5%, respectively). No significant genotype interactions with site or management system were found; however, the genotype by individual environment (GxE) interaction was significant (Table S3), and when analysing each system separately, the interactions in an organic system GxYxS ($p = 0.02$) and in a conventional system GxY ($p = 0.01$) were significant. Heterogeneous material provided a higher protein content than homogeneous varieties (significant differences between the diversity in all combinations) (Table 2). All the CB populations except MIX DK ranked higher than three out of the four CB checks; however, for individual genotype comparisons, the difference was significant only between the lowest protein variety, Anakin, and all CCPs and mixtures (Table S6). The protein contents of the CB populations did not significantly differ from the best yielding local check Rubiola. The HB populations ranked higher than the HB check Irbe, with significant differences between the two populations. No significant differences between groups of CCPs and mixtures were found across all environments; however, CCPs were, on average, superior over conventional environments (Table 2). To compare individual pairs, CCP-3 surpassed MIX 3 by 0.57% ($p = 0.03$) and CCP-6, also derived from male-sterile crosses, ranked higher than the respective mixture (a difference of 0.33%); therefore, a higher protein content could be associated with the effect of having male-sterile parents.

**Table 3.** Yield stability and adaptability indices: average yield (t ha$^{-1}$), coefficient of regression (b), deviation from regression (s$^2_{dj}$), and number of rankings in the upper third part (TOP) in organic (O), conventional (C), and combined (O+C) environments.

| Subjects | Yield | b ** | s$^2_{dj}$ | Yield | b ** | s$^2_{dj}$ | Yield | b | *p*-Value | TOP | | |
|---|---|---|---|---|---|---|---|---|---|---|---|---|
| | O+C | | | O | | | C | | | O+C | O | C |
| | *n* = 12 | | | *n* = 9 | | | *n* = 3 | | | *n* = 12 | *n* = 9 | *n* = 3 |
| CCP-Mirga | 2.92 | 1.08 | 0.05 | 2.44 | 1.18 ^ | 0.04 | 4.35 | 0.91 | 0.04 | 10 | 7 | 3 |
| MIX Mirga | 2.88 | 1.07 ^ | 0.02 | 2.45 | 1.15 ^ | 0.02 | 4.17 | 1.06 | 0.03 | 11 | 8 | 3 |
| CCP-3 | 2.59 | 0.94 | 0.05 | 2.15 | 0.92 | 0.04 | 3.90 | 0.90 | >0.05 | 4 | 3 | 1 |
| MIX 3 | 2.72 | 1.02 | 0.03 | 2.30 | 1.04 | 0.02 | 3.96 | 1.13 | 0.04 | 8 | 7 | 1 |
| CCP-4 | 2.86 | 1.06 | 0.04 | 2.41 | 1.14 | 0.04 | 4.22 | 0.97 | >0.05 | 9 | 7 | 2 |
| MIX 4 | 2.74 | 1.11 ^ | 0.02 | 2.26 | 1.19 ^ | 0.02 | 4.16 | 1.03 | 0.02 | 8 | 5 | 3 |
| CCP-5 | 2.78 | 0.96 | 0.06 | 2.39 | 1.06 | 0.07 | 3.94 | 0.90 | 0.02 | 6 | 5 | 1 |
| MIX 5 | 2.50 | 0.96 | 0.04 | 2.07 | 0.98 | 0.03 | 3.77 | 0.94 | >0.05 | 1 | 1 | 0 |
| CCP-6 | 2.67 | 1.03 | 0.02 | 2.19 | 0.99 | 0.02 | 4.09 | 1.01 | 0.01 | 6 | 4 | 2 |
| MIX 6 | 2.71 | 1.13 ^ | 0.02 | 2.23 | 1.13 ^ | 0.03 | 4.17 | 1.16 | >0.05 | 6 | 4 | 2 |
| CCP-7 | 2.70 | 0.87 * | 0.04 | 2.31 | 0.92 | 0.05 | 3.86 | 0.76 | 0.05 | 5 | 4 | 1 |
| MIX 7 | 2.60 | 1.03 | 0.05 | 2.17 | 1.20 ^ | 0.01 | 3.88 | 0.82 | >0.05 | 5 | 4 | 1 |
| MIX DK | 2.69 | 1.01 | 0.09 | 2.31 | 1.22^ | 0.05 | 3.85 | 0.87 | 0.04 | 6 | 6 | 0 |
| Rubiola | 2.80 | 1.20 ^ | 0.06 | 2.25 | 1.09 | 0.05 | 4.46 | 1.29 | >0.05 | 5 | 3 | 2 |
| Rasa | 2.38 | 1.02 | 0.06 | 2.01 | 1.13 | 0.04 | 3.51 | 1.12 | >0.05 | 1 | 1 | 0 |
| Abava | 2.45 | 0.98 | 0.04 | 1.97 | 0.86 * | 0.03 | 3.87 | 1.02 | >0.05 | 2 | 2 | 0 |
| Anakin | 2.69 | 1.29 ^ | 0.12 | 2.08 | 1.15 ^ | 0.04 | 4.52 | 1.37 | >0.05 | 3 | 1 | 2 |
| CCP-2HB | 1.76 | 0.89 * | 0.02 | 1.35 | 0.86 * | 0.03 | 2.98 | 0.90 | 0.02 | 0 | 0 | 0 |
| MIX 2 | 1.81 | 0.90 | 0.05 | 1.36 | 0.73 * | 0.02 | 3.15 | 1.01 | >0.05 | 0 | 0 | 0 |
| CCP-3HB | 2.08 | 0.93 | 0.05 | 1.63 | 0.79 * | 0.01 | 3.43 | 1.00 | >0.05 | 0 | 0 | 0 |
| CCP-5HB | 1.80 | 0.81 * | 0.05 | 1.47 | 0.79 * | 0.04 | 2.81 | 0.91 | >0.05 | 0 | 0 | 0 |
| CCP-7HB | 1.86 | 0.73 * | 0.06 | 1.55 | 0.75 * | 0.08 | 2.77 | 0.78 | >0.05 | 0 | 0 | 0 |
| MIX DK HB | 1.84 | 0.81 * | 0.05 | 1.47 | 0.71 * | 0.05 | 2.94 | 0.92 | 0.02 | 0 | 0 | 0 |
| Irbe | 2.18 | 1.14 ^ | 0.10 | 1.64 | 1.02 | 0.11 | 3.78 | 1.22 | 0.03 | 0 | 0 | 0 |

^ b value significantly above 1, * b value significantly below 1 (*p* = 0.05); ** *p* < 0.01.

The average TGW values in all environments ranged from 49.7 g for CB check Anakin to 38.4 g for HB check Irbe. Similar to protein content, the GxS, GxM, and GxY interactions were not significant, whereas the GxE and GxYxS interactions were significant (Table S3). Heterogeneous materials provided higher TGW values than the check varieties (Table 2). The values of CCP-3, MIX 3, and CCP Mirga did not significantly differ from the most superior check, Anakin; all Latvian CB CCPs performed similarly to check Rubiola, and all HB CCPs significantly surpassed check Irbe (Table S6). Significant differences between the CCPs and their respective mixture groups (Table 2), and between two individual pairs, CCP-5 > MIX 5 (*p* = 0.001) and CCP-6 > MIX 6 (*p* = 0.03), were found.

### 3.4. Nitrogen Use Efficiency (NUE)

Strong environmental effects were observed, causing statistically significant variations in NUE, depending on the year and location. Similarly, significant variations due to the year and location affected both NUE components, NUpE and NutE; the average values

were significantly higher in 2020 as well as in StOrg. The G×E and GxY interactions were also significant but the GxS interaction was not (Table S3).

The NUE values of the CB heterogeneous materials and homogeneous check varieties did not differ significantly; however, higher values were found in heterogeneous materials, and the CCP group was superior. For NUtE, significant differences ($p < 0.01$) and higher average values were found in the heterogeneous CB materials, including CCPs and mixtures, compared with the homogeneous varieties. Overall, for NUE and its components, there were no significant differences between CCPs and mixtures for both covered and hulless barley (Table 2).

Significant differences ($p < 0.001$) were found for the average values of NUE and NUtE between the CB and HB materials, and CB performed best. With HB, significant differences in NUE between the homogenous check and heterogeneous materials were observed, with higher NUE values for the variety Irbe; however, only CCP-2HB, out of the five populations, was significantly lower (Table S6). For this type of barley, a partial eta squared value ($\eta^2$) measuring the effect sizes of different variables by ANOVA showed that for NUE the proportion of variance explained by genotype was 10.2% ($p < 0.05$). CB populations CCP-4 and CCP-5 ranked, on average, highest in NUpE values across the five environments, resulting in the highest NUE value as well. For both types of barley, a significant variation ($p < 0.01$) between the subjects was found only in NUtE, where the proportion of variance explained by genotype was 21.3%. The highest NUtE among the CB groups was observed in heterogeneous materials, with the highest values being for MIX DK and MIX 3 (significantly different from the poorest performing check Abava), but CCP-3HB was the highest among HB. Comparing NUE and its component average values across all environments of the CCP group with the respective mixtures, the differences were not significant (Table 2); however, for the individual pair of CCP-4/MIX 4, the population was superior according to the NUpE value ($p = 0.018$). In addition, for CCP-5, there was a positive trend for all three parameters when compared with MIX 5.

### 3.5. Ability to Suppress Weeds

The weed suppression ability and crop ground cover were significantly affected by the factors Y, S, and E. On average, the highest crop ground cover was observed in 2019, but the highest weed suppression performance was registered in 2021. The highest crop ground cover was at site StOrg, which also provided the highest yields; however, higher weed suppression was observed at PrOrg (Table S3). In addition, the grain type had a significant effect on the weed suppression ability in crop GS 31–39 and GS 87–92 as well as on crop ground cover at both growth stages. A significant YxS interaction was found for all traits related to weed competitiveness. The GxE interaction only significantly affected crop ground cover.

No significant differences in weed competitiveness were found between CCPs and varieties or mixtures (Table 4). However, trends were observed. Overall, CCPs had higher weed competitiveness than the check varieties, though this was lower than that of the mixtures of parents. When comparing the performance of CB and HB, CB had both higher crop ground cover and higher weed suppression ability, which could have been related to the relatively poorer emergence and crop establishment of HB. The populations CCP-6 and CCP-7 were among the top five in terms of both crop ground cover and weed suppression ability (Table S7). CCP-3HB was the best among the HB subjects. For CB, the results by diversity groups showed that those with relatively higher weed competitiveness had heterogeneous CB material. During the earlier crop growth stages, CB CCPs had higher crop ground cover and weed suppression ability than the mixtures and the check varieties. For HB, CCPs had higher crop ground cover and higher weed suppression ability throughout the growing season. At earlier crop growth stages, Irbe had a higher average weed suppression ability.

**Table 4.** Differences between subjects grouped by their diversity level (*p*-value) and average values of weed competitiveness and disease severity traits.

| Diversity Groups | Crop Ground Cover, % | | Weed Suppression Ability, % | | | Net Blotch, AUDPC | Powdery Mildew, AUDPC | Loose Smut, Plants Per m² | Covered Smut, Plants Per m² |
|---|---|---|---|---|---|---|---|---|---|
| | GS 25–29 | GS 29–31 | GS 31–39 | GS 59–65 | GS 87–92 | | | | |
| **Heterogeneous vs. Homogeneous** | 0.540 | 0.588 | 0.486 | 0.367 | 0.407 | <0.001 | 0.062 | 0.523 | <0.001 |
| Heterogeneous CB | 40.3 | 54.5 | 37.6 | 46.1 | 50.0 | 4.0 a | 20.2 | 0.32 | 0.10 b |
| Homogeneous CB | 39.3 | 53.4 | 35.8 | 43.4 | 47.7 | 75.4 b | 15.1 | 0.35 | 0.01 a |
| **CCP vs. MIX vs. checks (CB)** | 0.445 | 0.650 | 0.728 | 0.663 | 0.679 | <0.001 | 0.173 | 0.078 | <0.001 |
| CCPs CB | 41.2 | 55.2 | 38.1 | 46.2 | 49.6 | 42.1 a | 20.1 | 0.37 | 0.05 a |
| Mixtures CB | 39.3 | 53.7 | 37.1 | 45.9 | 50.4 | 39.7 a | 20.5 | 0.26 | 0.16 b |
| Checks CB | 39.3 | 53.4 | 35.8 | 43.4 | 47.7 | 75.4 b | 15.1 | 0.35 | 0.01 a |
| **CCP vs. MIX vs. checks (HB)** | 0.580 | 0.662 | 0.767 | 0.925 | 0.904 | 0.376 | 0.032 | 0.129 | <0.001 |
| CCPs HB | 34.3 | 48.6 | 32.2 | 41.9 | 44.0 | 38.8 | 19.3 b | 0.30 | 2.82 b |
| Mixture HB | 32.3 | 47.1 | 28.8 | 39.8 | 41.8 | 42.6 | 13.5 ab | 0.18 | 2.72 b |
| Check HB | 31.0 | 44.9 | 32.6 | 42.0 | 43.0 | 44.9 | 6.9 a | 0.27 | 0.01 a |
| **CCP vs. MIX (pairs)** | 0.378 | 0.421 | 0.555 | 0.780 | 0.807 | 0.317 | 0.721 | 0.215 | 0.452 |
| CCPs | 39.7 | 54.3 | 37.4 | 45.8 | 48.6 | 37.9 | 20.33 | 0.29 | 0.65 |
| Mixtures | 38.3 | 52.7 | 35.9 | 45.1 | 49.2 | 40.1 | 19.47 | 0.25 | 0.53 |

O—organic, C—conventional, O+C—both farming systems; Bold—the row shows *p*-values of comparison between the respective groups; trait values marked with different letters indicate significant differences between the groups (*p* = 0.05); CB—covered barley, HB—hulless barley.

### 3.6. Disease Severity

Net blotch (*Pyrenophora teres*) was the most severe leaf disease during the study period. The disease severity assessed according to AUDPC values was significantly higher in Priekuli under an organic management system (average AUDPC values for O and C management, 60.2 and 37.3, respectively) and especially in 2020 (AUDPC at PrOrg 78.2), when the weather was cooler and with more moisture compared with the other years (Table S3). Heterogeneous CB materials were significantly less infected than the homogeneous check varieties (Table 4). All populations and mixtures except the Danish CB population ranked higher than all the checks and showed significantly less infection than the three most susceptible checks, Anakin, Abava, and Rasa (Figure 4). Among the CCPs and their mixtures of parents, no significant differences were found; however, CCPs had a slight advantage in five out of seven cases.

In terms of the severity of powdery mildew (*Blumeria graminis*; average AUDPC values for individual environments from 4.9 for PrConv 2021 to 48.2 for PrConv 2019), there were no significant differences between the average values, regardless of the crop management system (Table S3). However, in 2021, mildew severity was significantly higher in the organic than in the conventional management system (*p* = 0.05), which was poorly developed due to the effects of abiotic stress. No significant differences between the groups of homogenous and heterogeneous materials were found, with the exception of HB, where the CCPs were more infected than the check variety. Three checks (Anakin, Rasa, and Irbe) ranked as the most resistant in the experiment, whereas Rubiola and Abava were among the most susceptible; the bulk of the populations and mixtures ranked in between those two extremes (Figure S4). Three CCPs and one mixture were significantly more infected than the fully resistant check Anakin. A total of 15 heterogeneous subjects performed better than the moderately susceptible Rubiola, whereas only the most infected MIX 5 was significantly higher. No significant differences between the CCPs and mixtures were found. The average AUDPC value for the CCPs was slightly higher than for mixtures. This trend

was particularly evident in CCP-Mirga, CCP-3, and CCP-4, whereas MIX 5 was 63% more infected than CCP-5. The negative correlation between net blotch and powdery mildew AUDPC values (r = −0.199, *p* < 0.01) was attributed to the contrasting resistances to both diseases in the check varieties and the differences in disease pressure during the natural selection of CCPs prior to this experiment. In addition, there are few major resistance genes specifically for powdery mildew in the parental material, in contrast to quantitative net blotch resistance, which is more common in parents and appears to be a trait that is more effectively improved during natural selection.

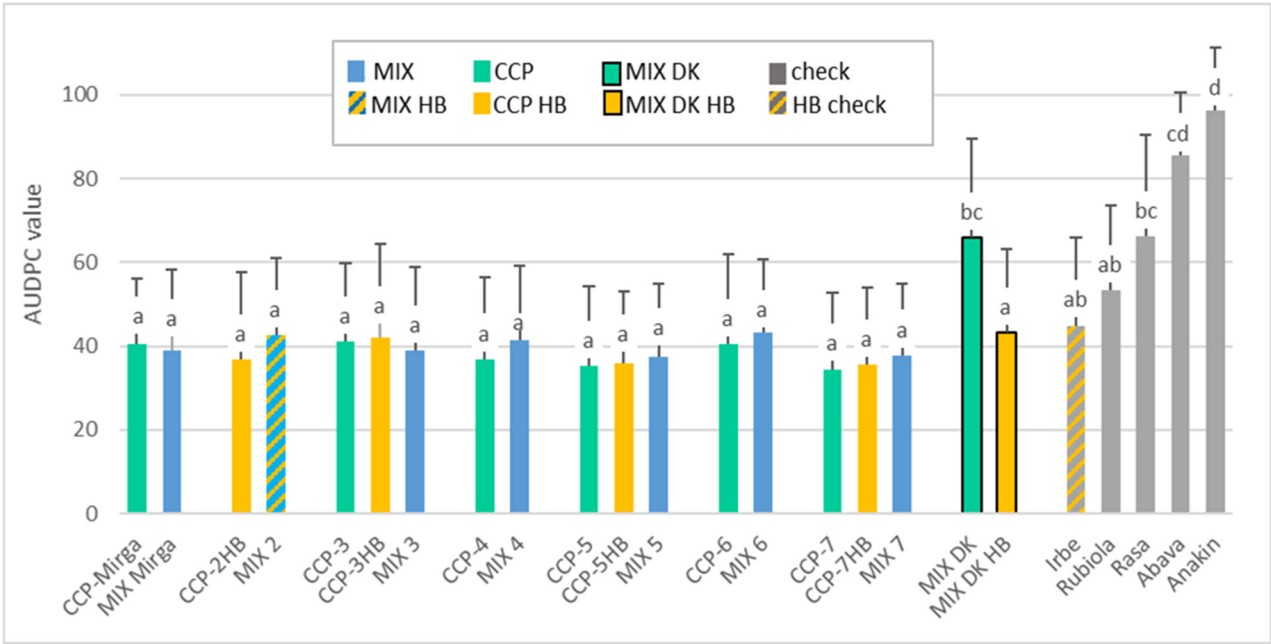

**Figure 4.** Average net blotch severity in two organic and three conventional environments. Error bars indicate standard deviations. Values marked with different letters indicate significant differences (*p* = 0.05). MIX—mixture, CCP—composite cross population, HB—hulless barley, DK—Denmark (country of origin), check—check variety.

Barley loose smut (*Ustilago nuda*) infection was significantly higher at PrConv than at the organic sites, and it was particularly high in 2021. No significant differences were found when comparing heterogeneous materials vs. homogeneous varieties (Table 4); however, there were significant differences between the Latvian checks and Danish check Anakin (Figure S5). Rubiola and Rasa ranked as the most resistant, Anakin was the most susceptible, and the populations and mixtures ranged in between the two groups. All heterogeneous subjects showed significantly better resistance than the most susceptible check, Anakin, with the exception of MIX DK, and in most cases, it was not significantly lower than Rubiola and Rasa (exceptions: CCP Mirga, MIX Mirga, and both Danish populations). Mirga includes two conventional varieties that are highly susceptible to loose smut. Most CCPs where loose smut resistance was a breeding goal (see Table S1) had comparatively higher resistance, except for CCP-5, whereas CCP-5HB was the least infected population. Heterogeneous HB material, with the exception of the Danish HB population, did not significantly differ from the HB check Irbe. No differences between populations and mixtures were found. For CCP-3, CCP-7, and CCP-7HB, a trend of being less infected than the respective mixture was observed and may have been related to an additional resistance derived by excluding male-sterile parents from the mixtures.

The number of infected plants with the second most severe seed-borne disease, barley-covered smut (*Ustilago hordei*), was, on average, significantly higher at the conventional site in 2019 (Table S3). For CB subjects, no significant differences between the checks

and heterogeneous materials were observed; however, the average heterogeneous and homogeneous groups differed significantly, with mixtures being the most infected (Table 4). This could have been caused by a few susceptible HB parents being included in MIX 5 and MIX 7. There were significant differences between HB and CB ($p < 0.001$), with the average numbers of infected plants being 2.40 and 0.08 per m$^2$, respectively. The infection of HB check Irbe, with 0.01 covered smut plants per m$^2$, was similar to that of the CB checks, whereas the infection of heterogeneous HB subjects ranked the lowest at 0.9–4.4 plants per m$^2$, with four of them being significantly lower. No difference between the average values of CCP and mixture pairs was found; however, the individual comparisons showed significant advantages for CCP-3, CCP-5, and CCP-7 compared to the respective mixtures.

## 4. Discussion

### 4.1. Heterogeneous Material Versus Homogenous Check Varieties

One of the objectives for our study was to compare the performance of heterogeneous spring barley material, primarily composite cross populations, to a set of homogeneous check varieties currently grown on farms in Latvia according to traits with specific importance for organic crop management systems.

Our results showed significant advantages for yields in organic and conventional stress environments, yield stability in contrasting environments, NUtE, protein content, 1000-grain weight, and net blotch severity as well as some positive trends for NUE, NUpE, and competitiveness against weeds. The main recognised disadvantage was the high susceptibility of HB populations to covered smut.

The superiority of evolutionary populations over uniform varieties in terms of their yields was reported in previous studies [8], especially when grown organically [16,22,23], in low-input environments [24], and under abiotic stress conditions [7,14]. In our study, one of the local CCPs ranked highest in each organic environment as well as in the conventional stress environments (with the exception of StOrg in 2020 with MIX Mirga). Our CCP Mirga (originally CCP-1), developed for a high yield, was identified to be productive in organic farming systems in our previous research studies [41,42]. The current study showed that CCP-Mirga was the highest ranked population across all 12 environments. However, across nine organic environments, the newer CCP-4, which combined a higher number of parents better suited for organic farming and more resistant to diseases, performed similarly and had advantages in other traits. The population involving the highest diversity (32 parents), CCP-7, was superior in the two lowest yield environments (PrOrg and StOrgF in 2019), emphasising the value of higher diversity under poor conditions. In contrast, the commercial varieties were sensitive to poor cultivation conditions [23], which was supported by the lower ranking of check varieties in most of the organic and conventional stress environments. Merrick et al. reported the superiority of evolutionary populations in terms of yield in the environments where they had been cultivated for five seasons [25]. In our experiment, local CCPs were grown in PrOrg and PrConv for several years prior to our study, depending on the time of creation (1–6 seasons, see Table S1), so their adaptation times were different. Populations CCP-5, CCP-7, and CCP-4 performed better at PrOrg, whereas the Danish population MIX DK, with just one year of prior adaptation, ranked comparatively higher at both StOrg and StOrgF (Figure S2).

The heterogeneous populations were reported to be highly stable in yield across environments compared with the homogeneous controls and parents [8,24,25]. Similarly, the CB populations were typically stable across all 12 environments, whereas most check varieties provided adaptation to favourable conventional conditions. In the research on variety mixtures, stability was found to increase with component diversity [43], and the CCPs with broader genetic bases were reported to have the greatest stability [32]. However, we found that CCP-7, which contained the highest number of parents, adapted to unfavourable low-yield environments. This was a valuable observation related to the diversity of organic farm practices and environments. The yield stability of the populations is attributed to the increased ability to compensate when parents of different adaptabilities are included [25,44].

For two populations, CCP-Mirga and Danish-origin MIX DK, we observed good adaptability to favourable environments under the organic crop management system. This is likely due to the favourable climatic and soil conditions in the country of origin for MIX DK and the comparatively higher number of conventionally bred parents used in CCP-Mirga. We found that adaptability could be influenced by several factors, including the parental gene pool, diversity level, and prior cultivation environments.

The CCPs exhibited better performance for plant traits related to weed suppression, such as early crop ground cover, leaf area index, plant height, and biomass wet weight, than their parents grown in monocultures [8]. However, Bocci et al. did not find evidence for improvements in terms of weed suppression, ground cover, or reduced weed density in these populations [22]. We also did not find significant differences for weed suppression and crop ground cover. At least seven heterogeneous subjects ranked higher than the best performing variety, Abava, in weed suppression for each of the three growth stages, and two populations were nearly aligned with Abava in terms of crop ground cover. Improvements in their competitive abilities were achieved by our newer populations (CCP-6, CCP-7, and CCP-5) since our previous study had reported significantly lower ground cover for our initially developed populations compared with Abava. The positive effects of natural selection on traits related to early vigour were identified in a study by Raggi et al. [31]. We also observed a slightly higher gain in developed populations over varieties when comparing competitive ability indices in the earlier growth stages. The better root adaptation in CCPs may account for their superior competitive abilities against weeds and nutrient use efficiency under organic conditions compared with modern varieties [15]. Most varieties were created in and/or for conventional crop management systems, whereas the natural selection employed by CCPs during their cultivation in organic environments over several years resulted in their improved adaptability to such conditions, especially for traits positively related to productivity. We also found significant advantages for populations and mixtures in NUE component N utilisation efficiency as well as a positive trend for N uptake efficiency.

In line with our results, Brumlop et al. reported a higher protein content in wheat CCPs than reference varieties. They also pointed out that quality traits are not subjected to natural selection [16]. We speculate that, as natural selection favours prevalence, the most productive plants may reduce their protein content since a negative correlation has been identified between yield and protein content. However, our data showed that this correlation was greater under conventional ($r = -0.505$, $p = 0.01$) than organic conditions ($r = -0.163$, $p = 0.01$) and, therefore, should affect protein content under organic cultivation to a lesser extent. The stability in protein content over time is correlated with genetic diversity in wheat [26], which has also shown different relationships within heterogeneous populations compared with traditional uniform varieties. CCP-3 derived from crosses with male-sterile mother lines had the highest protein content in the CB group, which may have been caused by the diverse gene pool of male-sterile parents and, to a lesser extent, by the presence of a small amount of sterile shrunken endosperm kernels in the harvest (approximately 1% in harvest 2021). Its TGW was also one of the highest. However, the average yield of CCP-3 ranked the lowest in CB populations in organic environments, which is in agreement with the results of Döring et al., where slightly lower yields were reported in wheat CCPs derived from male-sterile crosses than those of regular CCPs [8]. Male-sterile-cross-derived CCP-6 is derived from improved mother lines, which is likely the reason for its slightly higher yield but lower protein content. The TGWs were similar for the CB populations and check varieties, though further research is needed concerning kernel size uniformity, which is important in malt and food production. HB provided significantly higher protein content and lower TGW than CB ($p < 0.01$), which is in agreement with previous studies and related to the hull removal. High-protein HB populations may be suitable for specific feeds and food usage.

Positive indications for diverse materials and plant diseases have been reported [25]. However, Döring et al. did not find significant differences for *Septoria* in wheat CCPs [8].

Our results show that the gain was different for different diseases. For net blotch, which was the most widespread barley leaf disease during the population development and experimental stages, we found a significant advantage for heterogeneous materials over check varieties, but regarding the less widespread powdery mildew, no advantage could be identified. Complete mildew resistance could be possible in the presence of major resistance genes (e.g., *mlo11* in Anakin), but it would be difficult to achieve in CCPs with many different parents and would not favour diversity. Due to the lower infection rates, acquiring resistance against powdery mildew via natural selection is less effective compared with net blotch. In the case of seed-borne diseases, such as loose and covered smut, acquiring resistance via natural selection was also less effective. The best yielding CCP-Mirga was the most susceptible to loose smut and was excluded from growing on farms; we anticipate improving its resistance in future studies by crossing it with resistant breeding lines. HB is more susceptible to smut infections due to the removal of the hull and thus its protection. Our HB populations were highly infected by covered smut compared with the HB check and, consequently, could not be recommended for farm cultivation. However, we identified a significant decrease in the average number of infected plants over the years (Table S3), which may indicate a positive correlation with natural selection; this aspect will need additional research before any conclusions can be drawn.

### 4.2. Populations Versus Mixtures

Another objective of our study was to determine if there were any differences in the performances of seven composite cross populations and their respective mixtures developed from the same or similar parental genotypes. The mixtures were static in the first year of testing but became dynamic in the subsequent years.

Overall, we found only a few significant differences between both groups, such as the advantages of the CCPs in protein content under conventional crop management and in TGW. For grain yield, there were nonsignificant gains, mainly in low-yield and stress environments, whereas yield stability with significant differences in four pairs was the main advantage in favour of populations. Small nonsignificant advantages for net blotch, NUE, weed suppression, and crop ground cover were also noted.

Our results generally show good agreement with those of Döring et al., who also found minor and inconsistent differences in yield, stability, and grain quality. However, they reported obviously better early ground cover and plant height [8], whereas we observed only a slight tendency. Previous studies have reported heterogeneous populations to be clearly superior over mixtures of the same genetic background in terms of yield and stability [11,16,45,46]. Compared with the corresponding mixture, Brumlop et al. found more productive spikes, taller plants, and higher TGW values in winter wheat CCP, which also had better recovery from winterkill and was highly resilient with higher yields [16]. Similarly, we obtained a 9% yield gain in CCPs compared with mixtures under drought stress in conventional conditions and 10–19% in the lowest yield environments with an average yield less than 1.5 t ha$^{-1}$. Heterozygosity and segregation, along with a much larger number of distinct individuals, were present in populations in contrast to mixtures, and previous studies have suggested that these traits may provide better buffering against environmental changes [46]. In addition, natural selection favoured adaptation in CCPs for a longer period of time but was only able to have a minimal effect on mixtures. The significant advantage for CCP protein content under conventional but not organic systems could be explained by the previously mentioned differences in protein/yield relationships in both management systems since mixtures tended to have a better yield in conventional high-yield environments. A significant yield superiority of mixtures over CCPs in conventional systems has also been identified for wheat [8].

Döring et al. (2015) concluded that CCPs offer greater potential for rapid and dynamic responses to change and higher exploitation of plant diversity. To summarise, breeding for CCPs required more effort in exchange for only a few significant advantages over

mixtures. However, our results were in favour of CCPs, especially under low-yield and stress conditions.

### 4.3. NUE of Heterogeneous Populations in Organic Crop Management System: Novelty of Our Research

Increasing nitrogen use efficiency (NUE) can contribute to sustainable agriculture. There are three main factors that can influence the NUE of a crop or cropping system, such as genotype, agronomic practices, and the environment [47]. According to Moll et al. [37], enhanced NUE may result from the increased efficiency of N recovery from the soil (NUpE) and a higher efficiency of N utilisation for grain formation (NUtE). In our study, we had to consider that the yield results would be influenced by the limited supply of available nitrogen in organic farming since N availability is crucial for yield formation, especially in the early developmental stages [48]. Our research environments followed an arable organic farming model, where the nutrient level was maintained by growing green manure crops in rotations; therefore, the overall soil fertility (1.5–2.3%) and available soil N (1.2–1.8%) in the trial fields were low. Nevertheless, our results show that grain yield was significantly ($p < 0.01$) positively correlated with NUE (r = 0.456), NUpE (r = 0.289), and NUtE (r = 0.499). This implies that a genotype selected due to its high yield has the ability to extract comparatively higher amounts of N from the soil and to utilise it more efficiently.

A precondition for NUE genetic improvement is genetic variation [49]. Our results demonstrate significant phenotypic variation regarding N use, indicating that the selection of this trait is feasible. Furthermore, we found a trend that CCPs showed comparatively higher average NUE values than the homogeneous varieties. To the best of our knowledge, no previously published comparisons have addressed NUE and its components under organic farming conditions for heterogeneous and homogenous materials in spring barley. In the research conducted on three wheat CCPs developed in the United Kingdom, the differences among the populations in N uptake appeared small. However, when comparing the conventionally and organically cultivated populations as two groups, the latter had extracted approximately 6% more N than the former. Although this difference was only visible in the plant samples cut at the flowering stage but not in the grain and straw samples [50], we speculate that organic environments may favour the natural selection of NUE-related traits. This also appears to be suggested in our results. After years of natural selection, some populations showed an increasing trend in NUE traits compared with mixtures. A significant difference in NUpE for one CCP/mixture pair was found.

After reviewing the agronomic and physiological aspects of NUE in conventional and organic cereal-based production systems, Kubota et al. suggested that NUE may be improved through a combination of management practices and breeding strategies specific to the management system [48]. Therefore, another important task for organic plant breeders is to integrate genetic and phenotypic information on NUE and its related traits to develop distinct types of cultivars adapted to specific farming practices [51]. For instance, a wheat study using a hydroponic technique showed that high-baking-quality CCPs had larger root and shoot systems than high-yield CCPs, suggesting the former had advantages in extracting N during stem elongation and grain filling [21]. Concerning evolving plant populations such as CCPs, a major known advantage is their ability to adapt to stressful, variable, and unpredictable environments. Furthermore, within a population, a competitive ability for photosynthates may exist in the roots and shoots that could affect potential grain yield [52]. This indicates additional research is required to fully understand the N dynamics in organic systems to develop effective selection tools and define selection criteria for NUE improvement in different varieties, including CCPs.

Since NUE is determined by multiple genetic factors and heavily influenced by the environment, the identification of its genetic basis has been challenging [49,53]. The values of NUE, NUpE, and NUtE vary greatly between seasons, and the environment (i.e., the effect of soil N) has been shown to have a greater influence on the phenotypic variability than the genotype [49,54,55]. Similarly, in our research, there was significant variation

between trial sites and years, which indicates significant genotype by environmental (GxS and GxY) interactions. The environment was the important factor underpinning the overall phenotypic variance for NUE ($\eta^2$% = 65.7), NUpE ($\eta^2$% = 55.7), and NUtE ($\eta^2$% = 37.7). The contribution of the genotype as a factor was significant only for NUE and NUtE, and the effect size of this variable was comparatively small ($\eta^2$% = 6.7 and 2.2, respectively). High levels of GxE interactions suggest that future research should be conducted in these target environments, both to understand the factors that contribute to NUE or to breed varieties with high NUE. This is especially important in organic systems that are characterised by environmental heterogeneity. Conclusions drawn in a review on wheat [47], which may be applicable to barley as well, indicate that future plant breeding programs should focus on delivering more N-efficient varieties by modifying the N uptake, utilisation, and remobilisation. Furthermore, studies on the identification of traits that significantly contribute to NUE should address the specificities of certain environments. The breeding of and research on heterogeneous material could provide valuable input by exploring the mechanisms and effects of natural selection.

## 5. Conclusions

When comparing heterogeneous spring barley materials (populations and mixtures) to homogeneous check varieties, we found significant advantages for yield in organic and conventional stress environments, yield stability across contrasting environments, NUtE, protein content, 1000-grain weight, and net blotch severity as well as some positive trends for NUE, NUpE, and competitiveness against weeds. The main disadvantage found is the high susceptibility of hulless barley populations to covered smut. We suggest heterogeneous populations as valuable alternatives to traditional uniform varieties for organic as well as particularly poor cultivation environments.

The yields of CCPs were more stable across distinct environments than those of varieties and mixtures, and CCPs ranked higher for yield in organic and abiotic stress environments, including those that were conventionally managed.

We found significant differences between composite cross populations and mixtures of their parents, indicating advantages for CCPs in protein content and 1000-grain weight. In addition, there was a nonsignificant yield gain, mainly in low-yield and stress environments, higher yield stability in most cases, and minor positive trends for net blotch, NUE, weed suppression, and crop ground cover. Although multi-component mixtures, especially if cultivated dynamically, can provide a performance similar to CCPs of the same genetic background, when considering the greater potential of CCPs to evolve and adapt to particular environments, they have better advantages overall. Our results should enable the use of genetic diversity on organic farms and, in particular, should encourage greater yields and income from poor-yielding areas and unstable growth conditions.

**Supplementary Materials:** The following are available online at https://www.mdpi.com/article/10.3390/su14159697/s1, Table S1: Detailed information on the barley populations, varieties, and mixtures included in the experiment; Table S2: Soil and crop management characteristics at four trial sites (PrOrg, PrConv, StOrg, and StOrgF) during 2019–2021; Table S3: Genotype and environmental factor significance and average trait values across sites, crop management systems, and years; Figure S1: Spearman's rank correlation coefficients for average genotype yields (n = 24) among the sites (PrOrg, PrConv, StOrg, and StOrgF) and environments grouped according to average yield level (low: <2 t ha$^{-1}$; medium: 2–4 t ha$^{-1}$; and high: >4 t ha$^{-1}$), $p$ = 0.01; Figure S2: Ranking of subjects according to grain yield (t ha$^{-1}$) among organic sites (PrOrg, StOrg, and StOrgF, n = 3) and in conventional stress environments in 2021 (PrConv21). Error bars indicate standard deviations. Values marked with different letters indicate significant differences within CB and HB groups ($p$ = 0.05); Table S4: Average grain yield of diversity groups in individual environments (organic sites PrOrg, StOrg, and StOrgF and conventional site PrConv during 2019–2021) and environments grouped by average yield level (low: <2 t ha$^{-1}$; medium: 2–4 t ha$^{-1}$; and high: >4 t ha$^{-1}$); Table S5: Average values in diversity groups for yield stability and adaptability indices: average yield (t ha$^{-1}$), coefficient of regression (b), deviation from regression ($s^2_{dj}$), and number of rankings in the upper third part (TOP) in organic

(O), conventional (C), and combined (O+C) environments; Figure S3: Regression lines of selected CCP and respective mixture pairs for grain yield (t ha$^{-1}$) across 12 environments; Table S6: Average values over environments (n) for traits related to grain quality and N use efficiency (NUE); Table S7: Average values across environments (n) for traits related to competitive ability against weeds and disease severity; Figure S4: Average powdery mildew severity in two organic and three conventional environments; Figure S5: Barley loose smut and covered smut severity in 12 environments.

**Author Contributions:** Conceptualisation, L.L.; methodology, L.L., M.B. and D.P.; software, D.P.; validation, D.P., M.B. and L.L.; formal analysis, D.P.; investigation, L.L., D.P., M.B. and I.L.; data curation, L.L., D.P., M.B. and I.L.; writing—original draft preparation, L.L., M.B. and D.P.; writing—review and editing, L.L., M.B. and D.P.; visualisation, L.L.; supervision, L.L.; funding acquisition, L.L. All authors have read and agreed to the published version of the manuscript.

**Funding:** This research was funded by the Latvian Council of Science, grant number lzp-2018/1-0404, acronym FLPP-2018-1.

**Institutional Review Board Statement:** Not applicable.

**Informed Consent Statement:** Not applicable.

**Data Availability Statement:** The data presented in this study are available in the Supplementary Materials.

**Acknowledgments:** The authors thank Anders Borgen for providing the seed from his barley populations and Duane Falk for providing the initial male-sterile barley material for crossing.

**Conflicts of Interest:** The authors declare no conflict of interest.

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
