# Peer review of "Agronomic Performance of Heterogeneous Spring Barley Populations Compared with Mixtures of Their Parents and Homogeneous Varieties"

_sustainability, doi:10.3390/su14159697_

Round 1

Reviewer 1 Report

The introduction of the paper could be more succinct (lean).

The objective is very long and comprehensive. Could synthesize more!

Along with the objective in the sequence, they already talk about results. It got weird. Maybe that part should be relocated for discussion!

The objective of this study was to compare the yield, stability, weed suppression, nitrogen use efficiency, incidence of disease, and other essential traits in spring barley composite cross-populations (CCPs) to that of mixtures of their parental genotypes and currently grown homogeneous varieties to determine whether the CCPs are better suited to organic farming practices and abiotic stress conditions. Our results showed that CCPs are superior in most of the investigated traits compared with varieties, and even  possesses a few advantages over mixtures representing the same genetic background.

Author Response

Reviewers comments and suggestions

Reply

The introduction of the paper could be more succinct (lean).

The objective is very long and comprehensive. Could synthesize more!

Along with the objective in the sequence, they already talk about results. It got weird. Maybe that part should be relocated for discussion!

Introduction was shortened by omitting information not directly connected to our research.

Objective was shortened.

A sentence summarizing main conclusions was included in Introduction after the objectives as it was suggested in the journal template: “briefly mention the main aim of the work and highlight the principal conclusions”. We also considered it weird and now deleted that sentence.

Reviewer 2 Report

The topic covered by the manuscript appears appropriate and current.

The manuscript has to be formatted acording to journal requirements.

Abstract and conclusions have to be re-consider. What is the main motivation of the study? 

Future research of this study is presented well-structured but please limitations should be discussed detail.Referencing and in-text citation styles need to be corrected. The clarity of figures need to be revised.

I think it is appropriate to insert a short section focused on the economic aspect that this new approach entails

Author Response

The manuscript has to be formatted acording to journal requirements.

Changes in formatting done but manuscript not yet transferred to the journal template since it would change the page numbering for other reviewers to check the corrections. After consulting the editor I found out that final formatting will be done by the journal.

Abstract and conclusions have to be re-consider. What is the main motivation of the study?

Some supplements and rephrasing in Abstract and Conclusions were done.

  The main practical motivation was to find out the agronomical value of heterogeneous populations if compared to existing varieties and to provide background for promotion of them. Comparison between CCPs and the respective mixtures can be essential from the breeders point of view (is crossing followed by natural selection be more beneficial than just mixing of parents)

Future research of this study is presented well-structured but please limitations should be discussed detail.

Our aim was to concentrate on agronomic performance, we could unfortunately not cover all the aspects of heterogeneous material in the discussion, not directly related to our topic.

If the reviewer means the limitations connected to growing on farms, we did mention the new EU regulation allowing marketing of HM seed. There is still no experience known besides the EC Temporary experiment (2017-2021). But we aimed to attract the interest of growers so they choose to try growing HM. Another limitation could be also the luck of plant breeders rights, how breeding for HM can be financed.

Referencing and in-text citation styles need to be corrected.

Corrections in references were done. However, citation style and in-text citations seems generally to correspond to the requirements.

The clarity of figures need to be revised.

Figure 2 – additional explanation added. Figure 3 was improved.  Figure 4 was supplemented and additional explanation added.

Please let us know if any further improvements are necessary!

I think it is appropriate to insert a short section focused on the economic aspect that this new approach entails

We did study agronomic performance and our current team does not include representatives from economical sciences. So we do not feel competent enough to discuss this aspect,   and it is not directly related to our research.

Thank you for this suggestion, we will consider to include this aspect in our future work!

Reviewer 3 Report

This manuscript titled in “Agronomic performance of heterogeneous spring barley populations compared with mixtures of their parents and homogeneous varieties” compares the yield, stability, weed suppression, nitrogen use efficiency, incidence of disease, and other essential traits in spring barley composite cross-populations (CCPs) to that of mixtures of their parental genotypes and currently grown homogeneous varieties. The results show that CCPs are superior in most of the investigated traits compared with varieties, and even possess a few advantages over mixtures representing the same genetic background. This work is meaningful and reasonably well executed. Before considering this manuscript for publication, the authors should consider the following points in any revision as follows:

1.      In this manuscript, the authors compare a number of factors that contribute to the occurrence of barley disease and other characteristics, but these conditions were studied to control the conditions of decline, and the authors should explain why are the CCPs better suited to organic farming practices.

2.      In the abstract, the authors should introduce the scientific problem in two short sentences and more about the research content of this manuscript.

3.      The introduction section can be shortened appropriately to add logic of this manuscript.

4.      What was the original work? Or what are the innovations?

5.      The authors mentioned “In the HB CCPs threshability was improved by only sowing seeds with naturally detached hulls. CCPs involving male sterility (CCP-3, CCP-6, and CCP-7) were subject of additional cross-pollination in F2, and starting from F3, negative selection of sterile seeds was performed to eliminate the sterile plants and their negative effect on grain yield.” Why do the authors choose sterile seeds? Did the authors consider to use standard seed to achieve comparative observation and better performance?

6.      Is it reasonable and effective to use visual method to evaluate weed field area?

7.      On lines 282, 285, 407, and 713, the authors should add periods to the sentences.

8.      What positive benefits can this study bring to agricultural development?

9.      The format of Figure 3 is messy and not reasonable enough.

10.  In Figure 2, the authors mentioned “Error bars indicate standard deviation.”. However, from the data, we can see that the data errors of Figure 2A and Figure 2B are very large compared with other groups of data. Can the authors explain why?

11.  On line 272-275, the authors mentioned “All CB CCPs ranked higher than the respective mixtures, on average, in low-yield-level environments and in PrOrg environments, with the exception of CCP-3, and all CCPs were exclusively found to perform better than their respective mixtures under conventional stress conditions.” Why is it better than its respective mixtures under conventional pressure? Please add relevant explanations.

12.  In Table 3, what do the authors mean by “a” and “b” appearing in the numbers? For example, “1.07a”, “1.15a”, “0.86b” in the data.

13.  In Table 4, please unify the data format of “Heterogeneous vs. Homogeneous”, “CCP vs. MIX vs. checks (CB)” and “CCP vs. MIX vs. checks (CB)”.

I will be happy to recommend for publication a revised version of the manuscript in Sustainability.

Author Response

1.      In this manuscript, the authors compare a number of factors that contribute to the occurrence of barley disease and other characteristics, but these conditions were studied to control the conditions of decline, and the authors should explain why are the CCPs better suited to organic farming practices.

We are not sure we understand what the reviewer means by “conditions of decline”. Please explain more in case our reply does not correspond!

Organic farming practices are more unstable and diverse between the farms and seasons and the compensation and complementarity mechanisms between the diverse plants within CCPs can improve the buffering capacity against instability and stress factors, including spread of diseases.

2.      In the abstract, the authors should introduce the scientific problem in two short sentences and more about the research content of this manuscript.

Changes in abstract done, however, the limited number of words allowed for only 1 additional sentence about scientific problem. We consider important to show main results/conclusions.

3.      The introduction section can be shortened appropriately to add logic of this manuscript.

Shortened

4.      What was the original work? Or what are the innovations?

We stressed one of innovations in cover letter and chapter of Discussion 4.3. NUE of heterogeneous populations in organic crop management system: novelty of our research

However, the main original work was testing of heterogeneous materials, particularly CCPs, in comparison to homogeneous varieties. Only a few results on barley are available (and not a lot on other crops as well) in this century and no results available from Baltic/Nordic climatic conditions

5.      The authors mentioned “In the HB CCPs threshability was improved by only sowing seeds with naturally detached hulls. CCPs involving male sterility (CCP-3, CCP-6, and CCP-7) were subject of additional cross-pollination in F2, and starting from F3, negative selection of sterile seeds was performed to eliminate the sterile plants and their negative effect on grain yield.” Why do the authors choose sterile seeds? Did the authors consider to use standard seed to achieve comparative observation and better performance

The aim of using male sterile parental lines for creation of CCPs is supplemented to manuscript Materials and methods section, namely “to make crossing technically faster and also to add additional diversity arising from those parents and potential natural pollination in F2”.

Elimination of male sterile seed is being done starting from F3 every year before sowing until the segregation is observed and the effect of sterility as such is considered as minor on the yield. However, there is the effect of genetic background arising from male sterile parents which can also affect yield and other traits.

To continue our research and investigate effects of male sterile parents, we are currently performing experiment in order to compare CCPs created by regular/standard crosses and crosses with male sterile lines using the same set of parents.

6.      Is it reasonable and effective to use visual method to evaluate weed field area?

We evaluated weeds three times in growing season (at three different crop growth stages) for three years. The visual method is suitable and allows to distinguish weed ground cover separately from crop ground cover, especially in plots with the crop.

7.      On lines 282, 285, 407, and 713, the authors should add periods to the sentences.

done

8.      What positive benefits can this study bring to agricultural development?

Introduction (line 111) and Conclusions (line 689) are supplemented to answer this.

Benefits can be promotion of growing of heterogeneous material in organic agriculture.

Our results should enable use of genetic diversity on organic farms and especially favour the yields and income from poor-yielding and growing unstable conditions.

9.      The format of Figure 3 is messy and not reasonable enough.

Some improvements done

10.  In Figure 2, the authors mentioned “Error bars indicate standard deviation.”. However, from the data, we can see that the data errors of Figure 2A and Figure 2B are very large compared with other groups of data. Can the authors explain why?

Figures 2A  and 2B show the average values over higher number and more diverse environments (in addition, for conventional conditions year 2021 was very different from 2019 and 2020) , therefore the deviations are larger than in the other sub-figures, where averages over smaller number of similar environments according to the yield level are shown.

11.  On line 272-275, the authors mentioned “All CB CCPs ranked higher than the respective mixtures, on average, in low-yield-level environments and in PrOrg environments, with the exception of CCP-3, and all CCPs were exclusively found to perform better than their respective mixtures under conventional stress conditions.” Why is it better than its respective mixtures under conventional pressure? Please add relevant explanations.

We added some additional explanation in Discussion (lines 603-606).

The conventional drought and late sowing stress conditions appeared to provide very similar yield ranking results to the organic conditions. Therefore we consider that the reasons for CCP advantages can be also the same, better buffering capacity provided by greater diversity and in smaller extent effect of natural selection (for conventional trial, were previous years were favourable, adaptation to such conditions is expected).

12.  In Table 3, what do the authors mean by “a” and “b” appearing in the numbers? For example, “1.07a”, “1.15a”, “0.86b” in the data.

The letters indicated significant differences of b value from value 1, showing the direction of adaptability.

The symbols and explanation under the table is improved now.

13.  In Table 4, please unify the data format of “Heterogeneous vs. Homogeneous”, “CCP vs. MIX vs. checks (CB)” and “CCP vs. MIX vs. checks (CB)”.

done

Round 2

Reviewer 3 Report

The authors have carefully addressed all the issues I raised previously. I recommend it for publication in Sustainability.